# Targeting Cancer Stem Cells in Triple-Negative Breast Cancer

**DOI:** 10.3390/cancers11070965

**Published:** 2019-07-09

**Authors:** So-Yeon Park, Jang-Hyun Choi, Jeong-Seok Nam

**Affiliations:** 1School of Life Sciences, Gwangju Institute of Science and Technology, Gwangju 61005, Korea; 2Cell Logistics Research Center, Gwangju Institute of Science and Technology, Gwangju 61005, Korea

**Keywords:** triple-negative breast cancer (TNBC), breast cancer stem cell (BCSC), self-renewal signaling pathways, metabolic plasticity

## Abstract

Triple-negative breast cancer (TNBC) is a highly aggressive form of breast cancer that lacks targeted therapy options, and patients diagnosed with TNBC have poorer outcomes than patients with other breast cancer subtypes. Emerging evidence suggests that breast cancer stem cells (BCSCs), which have tumor-initiating potential and possess self-renewal capacity, may be responsible for this poor outcome by promoting therapy resistance, metastasis, and recurrence. TNBC cells have been consistently reported to display cancer stem cell (CSC) signatures at functional, molecular, and transcriptional levels. In recent decades, CSC-targeting strategies have shown therapeutic effects on TNBC in multiple preclinical studies, and some of these strategies are currently being evaluated in clinical trials. Therefore, understanding CSC biology in TNBC has the potential to guide the discovery of novel therapeutic agents in the future. In this review, we focus on the self-renewal signaling pathways (SRSPs) that are aberrantly activated in TNBC cells and discuss the specific signaling components that are involved in the tumor-initiating potential of TNBC cells. Additionally, we describe the molecular mechanisms shared by both TNBC cells and CSCs, including metabolic plasticity, which enables TNBC cells to switch between metabolic pathways according to substrate availability to meet the energetic and biosynthetic demands for rapid growth and survival under harsh conditions. We highlight CSCs as potential key regulators driving the aggressiveness of TNBC. Thus, the manipulation of CSCs in TNBC can be a targeted therapeutic strategy for TNBC in the future.

## 1. Introduction

Breast cancer is a highly heterogeneous disease that displays diverse morphological features, variable responsiveness to different therapeutic options, and different clinical outcomes. In an attempt to treat patients more efficiently, breast cancer classifications have been developed. Triple-negative breast cancer (TNBC) is the most devastating form of breast cancer because of its aggressive nature. TNBC cells lack estrogen receptor (ER) and progesterone receptor (PR) expression and are negative for human epidermal growth factor receptor 2 (HER2) overexpression; thus, TNBC does not respond to hormonal or anti-HER2 therapies and currently lacks targeted therapy options. Moreover, patients with TNBC have a higher risk of early metastasis than patients with other subtypes of breast cancer, and TNBC patients with residual disease after chemotherapy have worse overall survival than do non-TNBC patients [1]

Cancer stem cells (CSCs) have been proposed as one of the determining factors contributing to tumor heterogeneity. Not all cancer cells have tumorigenic potential. Instead, a small subpopulation of cancer cells has the capacity for self-renewal and can recapitulate the heterogeneity of the original tumor. These distinct biological functions have been used to characterize CSCs in various types of cancer. For instance, in solid tumors, CSCs are capable of proliferating in anchorage-independent three-dimensional culture, thereby forming spheroids, while non-CSCs undergo anoikis-mediated cell death. Additionally, CSCs are highly tumorigenic; therefore, they can be serially transplanted into mice and induced to form tumors even at low numbers, while non-CSCs cannot. Tumor incidence has been used as the standard method for estimating CSC frequency. In 2003, Al-Hajj first identified that the cell fraction with the CD44+/CD24−/Lin- phenotype in breast cancer patient tissues could recapitulate tumor burden in mice [2]. Later, in 2007, Ginestier et al. discovered that a subpopulation of cells with high aldehyde dehydrogenase (ALDH) activity could initiate tumors in vivo and in vitro [3]. Since then, the CD44+/CD24− phenotype and high ALDH activity have become the “gold standard” signature for breast cancer stem cells (BCSCs). Accumulating evidence suggests that BCSCs with these phenotypes are responsible for cancer progression and metastasis as well as tumor initiation [4]. Although chemotherapy can eliminate the bulk of tumor cells, it fails to eliminate BCSCs, thereby making these cells the leading cause of therapy resistance and recurrence [5,6,7,8,9].

The CSC theory provides a different insight into the aggressive nature of TNBC. Histopathological analyses of breast cancer patient tissues have revealed that compared to non-TNBC tissues, TNBC tissues exhibit enriched ALDH1 and CD44+/CD24− expression signatures [10,11,12]. Additionally, TNBC cells have been reported to form mammalian spheroids (mammospheres) at a higher degree than non-TNBC cells [4,10,11,13,14]. At the transcriptional level, pluripotency-related transcription factors, such as SOX2 and MYC, are overexpressed in TNBC and display a positive correlation with poor prognosis [15,16]. These data suggest that the TNBC phenotype is highly similar to the CSC phenotype.

To validate this hypothesis, we investigated the gene expression profiles of TNBC patients. We obtained microarray data from five TNBC patients and fourteen non-TNBC patients from the gene expression omnibus (GEO) database (GSE27447, https://www.ncbi.nlm.nih.gov/geo), and we identified 1972 annotated genes that were differentially expressed between TNBC and non-TNBC (*p*-value < 5E-02, Appendix A). To link this TNBC gene set with specific biological functions, we performed gene set enrichment analysis (GSEA).-The detailed analytical method can be found in the Supplementary Methods. We discovered that the gene signature of TNBC cells was remarkably similar to that of mammary stem cells (Figure 1A). The up-regulated genes in mammary stem cells were also enriched in TNBC cells (Gene set ID: M2573), while the down-regulated genes were diminished (Gene set ID: M2574). Additionally, we conducted GSEA using a human breast cell line panel (GSE32474) to compare the genomic signature of TNBC cell lines (BT549, HS578T, and MDA-MB-231) with that of non-TNBC cell lines (MCF7 and T47D), and we identified a total of 653 annotated genes that were differentially expressed in TNBC (*p*-value < 1E-08, Appendix A). We repeatedly observed that the stem cell signature (Gene set ID: M1834) was significantly enriched in TNBC cell lines rather than in non-TNBC cell lines (Figure 1B). Our data are supported by the findings of a previous report by Yehiely and coworkers [17]. They classified the molecular subtypes of breast cancer by gene expression clustering and discovered a set of TNBC-related genes that included CSC-enriched genes responsible for self-renewal activity, such as c-KIT, TGF-β, α6 integrin subunit, and prion protein [17].

Furthermore, accumulating evidence indicates that epithelial to mesenchymal transition (EMT) is another connection between TNBC and CSC phenotypes. EMT is a cellular process that promotes the conversion of adherent epithelial cells into dissociated mesenchymal cells. During tumor progression, EMT is thought to be activated and to ultimately facilitate tumor cell migration through the basement membrane and subsequent invasion into adjacent tissues, followed by entry into the systemic circulation [18,19]. In particular, ectopic overexpression of EMT-promoting transcription factors, such as Snail, Twist, and Zeb1, promotes the transformation of mammary epithelial cells to BCSCs, suggesting that EMT may be a key process for the de novo generation of BCSCs [20,21]. EMT can be regulated by various stimuli, including oncogenic mutations and complex signaling networks that involve the tumor microenvironment; thus, EMT can serve as a key mechanism for balancing the cell state in which reprogramming to a CSC state enables the adaptation of cells to survive under harsh conditions [22,23,24,25,26,27,28]. Interestingly, TNBC cells highly express the key transcription factors that induce EMT, with consequent up-regulation of mesenchymal proteins and down-regulation of epithelial proteins. Moreover, the mesenchymal phenotype is responsible for the invasiveness and chemoresistance of TNBC cells [27,29,30,31,32]. Thus, the EMT signature, which is consistently observed in both CSCs and TNBC cells, provides evidence to support the similarities between the TNBC and CSC phenotypes.

Collectively, these data provide insights into the aggressiveness of TNBC by verifying the relationship between TNBC and CSC phenotypes. Therefore, understanding the role of CSCs may be promising for exploring novel targeted therapies for TNBC. In fact, many therapeutic strategies attenuating the CSC phenotype have shown inhibitory effects on tumor initiation, therapy resistance, and metastasis in TNBC cells. In this review, we introduce potential TNBC targets that have recently been identified and found to be involved in regulating CSC signatures. First, multiple self-renewal signaling pathways (SRSPs) have been reported to be aberrantly activated in TNBC cells, and inhibition of these signaling pathways has shown therapeutic effects on TNBC by reducing stemness. Second, CSCs are prone to metabolic reprogramming to adapt to harsh conditions, and the aggressive nature of TNBC is also dependent on this reprogramming process. In this review, we summarize the current knowledge of how TNBC harbors CSC signatures and suggest directions for future research.

## 2. Approaches for Targeting the Self-Renewal Process in TNBC Cells

Mammary stem cells regulate their self-renewal by multiple signaling pathways that are under strict regulation by intrinsic and extrinsic mechanisms, thus maintaining homeostasis in healthy tissues. Self-renewal activity is also essential for the maintenance and propagation of BCSCs; however, BCSCs escape strict regulation and rely on key dysregulated SRSPs, such as signal transducer and activator of transcription (STAT) signaling, Proto-oncogene tyrosine-protein kinase Src (SRC) signaling, and Wnt/β-catenin signaling, which lead to extensive cell propagation [33]. This aberrantly activated self-renewal capacity of CSCs is considered an early event in tumorigenesis and enables these cells to resist conventional chemotherapeutic agents, resulting in tumor recurrence. In the next section, we focus on the SRSPs that have been determined to be highly activated in TNBC relative to non-TNBC. We also introduce the specific signaling components that are overexpressed in TNBC along with the related drugs (Table 1).

### 2.1. STAT3

#### 2.1.1. STAT3 Signaling in BCSCs

Leukemia inhibitory factor (LIF) is a key cytokine maintaining the self-renewal and pluripotency of embryonic stem cells [18,34,35]. Among the many downstream signaling events involving the LIF receptor, the activation and DNA binding of STAT3 play a central role in transducing the functions of LIF. Additionally, Janus kinase (JAK)/STAT3 activation enables somatic cell reprogramming in the absence of otherwise essential pluripotency factors [36]. STAT3 activation also increases MYC mRNA transcription and prevents its proteasomal degradation by GSK3β-dependent phosphorylation at threonine 58 in embryonic stem cells [35].

In BCSCs, activation of STAT3 signaling has been implicated as a key mechanism for self-renewal regulation. Interleukin 6 (IL6) was discovered to induce the conversion of non-BCSCs into BCSCs by activating OCT4 transcription through STAT3, thereby increasing the self-renewal activity of breast cancer cells [37]. In addition, the binding of vascular endothelial growth factor (VEGF) to its receptor 2 (VEGFR2) induces STAT3 phosphorylation and promotes the binding of STAT3 to the SOX2 and MYC promoter regions for their transcriptional activation in breast cancer cells [38]. Through this mechanism, VEGF-mediated STAT3 activation increases the in vivo tumorigenic potential, mammosphere-forming efficiency, and ALDH activity of breast cancer cells [38].

#### 2.1.2. STAT3 Signaling Dysregulation in TNBC Cells

Various cytokines and growth factors, including IL6, IL8 and leptin, are linked to STAT3 activation in TNBC cells. Lehmann et al. performed a large-scale loss-of-function screening in TNBC cell lines and identified that a set of genes including IL6, PTGIS, HAS1, CXCL3, and PFKFB3 were required for the propagation of TNBC cells but not for the growth of non-TNBC cells [39]. In further validation experiments, the knockdown of IL6 signaling components, such as its receptor IL6RA and its downstream kinase JAK2, preferentially inhibited the growth of TNBC cells while having little or no effect on their non-TNBC counterparts [39]. Paralleling these results, Hartman et al. analyzed the genomic profiling of TNBC patient tumors in multiple databases and found a set of inflammation-related genes that were differentially expressed in TNBC cells [40]. Among them, the pro-inflammatory mediators IL6, IL8, and chemokine (C-X-C motif) ligand 1 (CXCL1) were critical for the anchorage-independent growth of TNBC cells yet exerted a minimum effect on non-TNBC counterparts. The results further showed that all twelve tested TNBC cell lines had higher levels of both mRNA transcripts and chemokine/cytokine release into the media than did the six ER-positive/luminal-like breast cancer cell lines via activation of nuclear factor-kappa B (NF-κB) signaling [40]. Blocking the IL6/IL8 autocrine loop decreased the survival and anti-apoptotic potential of TNBC cells and successfully sensitized TNBC cells to chemotherapy [40].

STAT family proteins are downstream transcription factors of the IL6/IL8/JAK2 signaling pathway. Among them, STAT3 is preferentially activated in its phosphorylated form in TNBC cell lines but not in non-TNBC cell lines, while STAT1 and STAT3 remain unaltered [39]. Moreover, the genomic signature of TNBC conferring STAT3 activation could be a predictive tool for poor prognosis in breast cancer patients [39]. Importantly, STAT3 overexpression was determined to be highly related to TNBC initiation, progression, metastasis, and chemotherapy resistance [41]. Leptin is also involved in the constitutive activation of the STAT3 signaling pathway in TNBC. Leptin and the long form of leptin receptor (LEPR) are enriched in breast cancer tissues and promote cell proliferation, migration, and angiogenesis [42]. Recently, studies have shown that the binding of leptin to its receptor, LEPR, initiates the activation of the JAK2/STAT3 signaling pathway, which further induces self-renewal and maintains the stem cell state of TNBC stem cells [43].

Through genomic profiling of BCSCs, Creighton et al. found that 493 genes were altered in BCSCs relative to non-BCSCs isolated from patient tissues [44]. Through shRNA screening for 493 genes, 13 genes were identified and defined as BCSC-regulating genes. Thus, hematological and neurological expression 1-like (HN1L) was discovered [45]. HN1L was overexpressed in breast cancer patient tissues compared with their normal counterparts, and strong HN1L expression had a positive correlation with poor prognosis only in TNBC patients, not in non-TNBC patients [45]. HN1L is required for TNBC self-renewal activity, migration and chemoresistance. HN1L enhances STAT3 activation by increasing the upstream receptors of STAT3, such as LEPR and fibroblast growth factor receptor 2 (FGFR2). HN1L expression is essential for the maintenance of SOX2, SOX9, and Kruppel-like factor 4 (KLF4) expression in TNBC cells [45].

STAT5 has also been reported as a key mechanism underlying TNBC resistance and CSC maintenance. Britschgi et al. observed that STAT5 activation promoted the resistance of TNBC cells to PI3K/mTOR inhibition [46]. Their study showed that IL8 secretion from breast cancer cells following PI3K/mTOR inhibition consistently enhanced JAK2/STAT5 phosphorylation but not STAT3 phosphorylation. Genetic silencing of STAT5 and JAK2 inhibition with NVP-BSK805 treatment sensitized TNBC cells to PI3K/mTOR inhibitors. Interestingly, TNBC cell lines (MDA-MB-468 and RAS-mutated MDA-MB-231) exhibited a more significant increase in JAK2/STAT5 phosphorylation and IL8 secretion than did non-TNBC cell lines (SK-BR3 and MCF7), and these cell lines possessed greater resistance to PI3K/mTOR inhibitors. Moreover, the same group determined that BCSCs could survive PI3K/mTOR inhibition and that the inhibition of IL8/JAK2/STAT5 signaling could significantly reduce the BCSC population as well as metastasis in TNBC xenograft models. Additionally, Bernaciak et al. confirmed that the loss of STAT5 reduced the migratory potential of some TNBC cell lines, such as MDA-MB-231 and BT-549 [47]. However, Walker et al. discovered that STAT5 activation increased the sensitivity of the TNBC cell line MDA-MB-468 cells to paclitaxel and vinorelbine [48]. Additionally, Sultan et al. and Tvorogove et al. validated that STAT5 activation promoted the reversal of EMT and the differentiation of both TNBC (BT-20 and MDA-MB-468) and non-TNBC (T47D) cell lines, thereby reducing their invasive and migratory potential [49,50,51]. Sultan et al. showed that JAK2 and STAT5 overexpression induced ER expression in the TNBC cell line BT-20 under 3D-organoid culture conditions [49]. These conflicting results in human breast cancer cell lines highlight the need for further investigation of STAT5 using clinical samples.

#### 2.1.3. STAT3 Signaling Inhibitors in Clinical Trials

There are various strategies for blocking STAT3 signaling. First, the ligand-receptor interaction can be blocked by antibodies. Second, STAT3 phosphorylation can be blocked by targeting the activity of its upstream kinases, such as JAK, or by interfering with the docking of STAT3 to its kinases. Third, the transcriptional activity of STAT3 can be blocked by preventing its dimerization, nuclear trafficking, and binding to DNA [52]. Merck-5 (JAK inhibitor I), which is an ATP-competitive inhibitor of JAK1/2/3 and TYK2, preferentially reduced the growth of TNBC cells without affecting non-TNBC cells [39]. Another ATP-competitive inhibitor of JAK1/JAK2, ruxolitinib, is currently in a phase II trial as a preoperative chemotherapy approach for inflammatory TNBC (NCT02876302) and in a phase I trial as a combinatorial drug for metastatic TNBC patients (NCT03012230). TTI-101 and OPB-51602 are orally bioavailable small molecules that specifically bind to the phospho-tyrosyl peptide-binding site within the SRC homology 2 (SH2) domain of STAT3. They inhibit JAK-mediated tyrosine phosphorylation, thus preventing the activation of STAT3. A phase I study of TTI-101 is currently recruiting patients with multiple types of cancer, including breast cancer (NCT03195699), and the phase I study OPB-51602 has been completed (NCT01184807). AZD9150 is a next-generation antisense nucleotide inhibitor for STAT3 and is currently in phase I and II clinical trials alone or in combination with chemotherapy in patients with advanced solid tumors (NCT03421353).

### 2.2. Proto-Oncogene Tyrosine-Protein Kinase Src *(SRC)*Signaling

#### 2.2.1. SRC Kinase Signaling in BCSCs

SRC kinase is a member of the nonreceptor tyrosine kinase family that plays key roles in regulating signal transduction by interacting with a diverse set of cell surface receptors under multiple cellular conditions [53]. Thakur et al. observed that SRC kinase was more highly phosphorylated in mammospheres than in monolayer-cultured breast cancer cells, suggesting that BCSCs exhibit high SRC kinase activity [54]. Moreover, SRC kinase inhibition by AZM475271 significantly reduced the self-renewal process and migratory potential of BCSCs [54]. Additionally, SRC kinase has been implicated in the chemo-/radioresistance of BCSCs. For instance, Kim et al. observed that SRC kinase signaling induced EMT in residual breast cancer cells after irradiation, thus increasing CSC signatures related to the CD44+/CD24- phenotype, chemoresistance, invasiveness, and migration [55,56], suggesting the importance of SRC kinase in therapy resistance.

#### 2.2.2. SRC Kinase Signaling Dysregulation in TNBC Cells

Recently, Tian et al. provided evidence to show that SRC kinase acts as a mediator of chemoresistance in TNBC cells [41]. TNBC cells that survived after paclitaxel treatment displayed a more enriched BCSC phenotype, such as CD44+/CD24- and high ALDH activity, and exhibited more enhanced mammosphere-forming capacity [41]. By screening multiple small-molecule inhibitors, the SRC inhibitor dasatinib was identified as the most potent reagent that reduced stemness signatures in these paclitaxel-resistant TNBC cells [41]. SRC activation drives the EMT signature in TNBC, and SRC inhibition by dasatinib treatment induces differentiation in TNBC cells, resulting in sensitization to paclitaxel [41]. Moreover, SRC kinase has been proposed as a key upstream protein for persistent STAT3 activation, which drives the robust outgrowth of breast cancer cells [57,58,59]. More recently, Lu et al. discovered a novel mechanism for the chemoresistance of TNBC cells in which the crosstalk between SRC and STAT3 signaling collectively contributes to the self-renewal process in BCSCs [60]. They discovered that TNBC cells highly expressed glutathione S-transferase omega 1 (GSTO1) after chemotherapy in a hypoxia-inducible factor 1α (HIF1α)-dependent manner. In turn, GSTO1 interacted with the type 1 ryanodine receptor (RYR1) and increased intracellular Ca2+ to trigger the activation of pyruvate kinase 2 (PYK2) as well as its downstream SRC kinase. SRC kinase activation increased KLF4 transcription through the direct binding of STAT3 to its promoter region, thus enhancing the self-renewal activity of TNBC cells [60].

#### 2.2.3. SRC Kinase Inhibitors in Clinical Trials

Dasatinib is an orally available dual tyrosine kinase inhibitor that binds to the ATP-binding site of SRC kinase. It was approved for use in patients with chronic myelogenous leukemia. In breast cancer, dasatinib treatment reduced the proliferation of TNBC cell lines in vitro as well as their tumorigenic potential in vivo [61]. Moreover, dasatinib treatment sensitized TNBC cells to paclitaxel by inducing apoptosis [41]. Dasatinib is currently in phase II trial in patients with breast cancer, including TNBC (NCT02720185). SKI-606 is an ATP-competitive dual inhibitor for BCR-ABL and SRC kinase. SKI-606 treatment reduced breast cancer invasion, growth, and metastasis both in vitro and in vivo [62,63]. A phase I study of SKI-606 is currently recruiting breast cancer patients (NCT03854903).

### 2.3. Wnt/β-Catenin Signaling

#### 2.3.1. Wnt/β-Catenin Signaling in BCSCs

The activation of Wnt/β-catenin signaling can be initiated by the binding of Wnt ligands to their receptor, the frizzled (FZD) family proteins, and their coreceptors, low-density lipoprotein receptor-related proteins (LRPs), resulting in Wnt-FZD-LRP complex formation [64]. Subsequently, β-catenin is released from the APC complex, including the AXIN-glycogen synthase kinase 3 (GSK3)-casein kinase 1 (CK1) complex (AXIN-GSK3-CK1), which phosphorylates and degrades β-catenin. This active β-catenin translocates into the nucleus to regulate gene transcription by binding with multiple transcription factors, including lymphoid enhancer-binding factor (LEF), T cell factor (TCF), and cAMP response element-binding protein (CREB)-binding protein (CBP) [65]. AXIN-GSK3-CK1 is recruited to the Wnt-FZD-LRP complex and then phosphorylates LRPs to stabilize the whole structure, thus amplifying Wnt/β-catenin signaling [64].

Wnt/β-catenin signaling has been implicated in embryonic development and homeostasis in adult stem cells [66,67]. The importance of Wnt/β-catenin signaling in BCSCs has also been validated in various experimental models [68]. Our research group previously determined that a set of genes involved in Wnt/β-catenin signaling, including WNT1, FZD1, TCF4, and LEF1, was up-regulated in BCSC-enriched mammospheres relative to that in more differentiated bulk cancer cells [69]. Additionally, when we isolated the cell subpopulation with high ALDH activity (ALDH-high), we detected a set of Wnt/β-catenin signaling proteins, such as LEF1, TCF4, and β-catenin that increased relative to that in the cell fraction with low ALDH activity [70]. Furthermore, we found that the ALDH-high activity population displayed more enhanced Wnt/β-catenin transcriptional activity than the ALDH-low activity population. When we treated breast cancer cells with a small-molecule Wnt/β-catenin inhibitor, BCSCs exhibited greater growth inhibition than did bulk tumor cells [69]. Moreover, Wnt/β-catenin inhibition sensitized BCSCs to docetaxel by reducing their self-renewal activities [69]. Genetic silencing of WNT1 in breast cancer cells also reduced the self-renewal activity and invasive potential of BCSCs, resulting in a reduction in tumorigenesis and metastasis in orthotropic xenografts in mice [70].

#### 2.3.2. Dysregulation of Wnt/β-Catenin Signaling in TNBC Cells

The nuclear accumulation of β-catenin, the signature of Wnt/β-catenin signaling activation, is evidently increased in TNBC compared with that in non-TNBC [71]. The nuclear accumulation of β-catenin has been shown to promote cell migration, colony formation, stem-like features and chemoresistance of TNBC cells and drive TNBC tumorigenesis in mouse cancer models, suggesting that Wnt/β-catenin signaling is a major driving force of TNBC tumorigenesis [72]. Green et al. compared the activity of the Wnt/β-catenin signal transduction pathway using the TCF reporter vector in multiple human breast cancer cell lines, including eight TNBC cell lines and seven non-TNBC cell lines. To this end, 7 of 8 TNBC cell lines (MDA-MB-231, HCC1395, Hs578T, BT549, CAL51, HCC1187, and MDA-MB-468) were determined to have higher activity of TCF-driven transcription in response to exogenous Wnt ligands than non-TNBC cell lines (T47D, HCC1419, SK-BR3, BT474, MCF7, SUM190PT, and SUM225) [73].

The overexpression of Wnt pathway genes, including CBP, FZDs, and LRPs, was observed in TNBC [74]. Ring et al. identified the amplification of CBP gene copy number, one of the crucial coactivators in β-catenin-driven transcription, in breast cancer patients [75]. Moreover, the overexpression of CBP transcripts was more enriched in TNBC subtypes than in non-TNBC subtypes [75]. Corda et al. systemically analyzed gene copy number variations in breast cancer patients and found FZD6 amplification in TNBC patients [76]. Histological assessment validated the overexpression of FZD6 protein in TNBC tissues [76]. Yang et al. compared the gene expression profiles in breast cancer patients through microarray analysis and identified the overexpression of FZD7 in TNBC [77]. Genetic silencing of FZD6 or FZD7 attenuated aggressive phenotypes of TNBC cells, such as motility, invasion, mammosphere formation, and in vivo tumorigenesis [76,77]. Yin and coworkers have recently linked FZD8-induced Wnt signaling to chemoresistance in TNBC cells [78]. The study showed that FZD8 expression increased in the residual cells after cisplatin and tumor necrosis factor-related apoptosis-inducing ligand (TRAIL) treatment in multiple TNBC cell lines and TNBC tumor xenografts. FZD8 depletion reduced β-catenin accumulation and increased chemotherapy-induced apoptosis in TNBC cells, indicating the up-regulation of FZD8 as a key mechanism in the development of chemoresistance in TNBC [78].

Wnt coreceptors, LRPs, have been shown to be up-regulated more frequently in TNBC, favoring cell proliferation, migration, invasion, and tumor growth [79,80,81]. LRP6 overexpression has been implicated in mammary tumorigenesis in both mice and humans in correlation with Wnt/β-catenin signaling activation [79,82]. Liu et al. recently performed a real-time PCR-based tissue array analysis in 41 breast cancer patients and identified an increase in LRP6 transcripts in breast cancer tissues compared with that in normal mammary tissues [79]. In particular, LRP6 expression increased in TNBC cell lines and patient tissues compared with that in the non-TNBC counterparts [79,80]. Genetic silencing of LRP6 diminished Wnt ligand-mediated signaling activation and abrogated in vivo tumorigenesis in TNBC xenografts [79]. LRP6 depletion further disrupted the invasion and migration of TNBC cells [80]. Lin et al. also identified that the mRNA expression level of LRP8 increased in TNBC patient tissues compared with that in tissue from patients with other subtypes of breast cancer, with a positive correlation with poorer prognosis [81]. LRP8 knockdown induced the transformation of TNBC cells from a mesenchymal state to a more differentiated epithelial state, thus reducing their self-renewal activity, invasion, and resistance to docetaxel [81].

Bin-Nun et al. recently identified a novel LRP6-interacting protein, protein tyrosine kinase 7 (PTK7) that physically interacts with LRP6 at its transmembrane domain and maintains LRP6 protein stability, thereby enhancing the Wnt/β-catenin pathway [83]. Genomic expression profiling of breast cancer cell lines and primary patient tumors by Gartenner et al. showed the preferential enrichment of PTK7 expression in TNBC and the necessity for PTK7 for the invasion and metastasis of TNBC cell lines [84]. Additionally, PTK7 protein levels assessed by immunohistological analysis were significantly related to the poor response of TNBC patients to anthracycline-therapy drugs [85]. Damelin et al. found that PTK7-expressing cells were more tumorigenic in vivo than PTK7-negative cells and that depleting the PTK7-expressing cells blocked tumor initiation and progression [86].

#### 2.3.3. Wnt/β-Catenin Signaling Inhibitors in Clinical Trials

There are several strategies to block Wnt/β-catenin signaling based on the location of the targets, including nuclear transcription, extracellular ligand secretion, and signaling receptors. ICG-001 was developed to interrupt the binding of β-catenin to CBP and successfully reduce the viability of the TNBC cell line MDA-MB-231 without affecting that of nontransformed MCF10A cells [75]. PRI-724 is a second-generation CBP inhibitor derived from ICG-001 [87]. Phase I and II clinical trials of PRI-724 have been completed in advanced pancreatic cancer patients (NCT01764477) and myeloid leukemia patients (NCT01606579), respectively.

To block Wnt ligand secretion, LGK-974 was developed as a small-molecule inhibitor for porcupine, which participates in Wnt ligand palmitoylation and enables extracellular secretion [88]. A phase I clinical trial of LGK-974 is currently recruiting patients with multiple solid cancer types, including TNBC [NCT01351103].

Gurney et al. developed the antagonistic Wnt pathway antibody OMP-18R5, which was initially isolated by its ability to bind to FZD7 but later discovered to bind to FZD2, FZD5, and FZD8 [89]. OMP-18R5 treatment reduced tumorigenesis in multiple types of human tumor xenografts and inhibited sphere-forming efficiency [89]. A phase I study of OMP-18R5 in combination with paclitaxel treatment has been completed in metastatic breast cancer (NCT01973309).

Damelin et al. developed a PTK7-targeted antibody-drug conjugate (PTK7-ADC), and treatment with PTK7-ADC depleted tumor-initiating cells and induced tumor regression in TNBC patient tissue-derived or TNBC cell line-derived xenografts [86]. Currently, a phase I study of PTK7-ADC as a combinatorial agent is recruiting patients with metastatic breast cancer and TNBC (NCT03243331).

### 2.4. Other Molecules Linked to TNBC Self-Renewal

#### 2.4.1. Connexin (CX)

Connexin family proteins are the subunits of gap junction (GJ) proteins and canonically function in GJ plaques at the interface of adjacent cells to facilitate direct cell-to-cell communication [90]. Thiagarajan et al. discovered that CX26 is differentially expressed between TNBC cells and non-TNBC and normal stem cells. CX26 displays an intracellular localization in TNBC cells, especially at the nuclear envelope, while CX26 is localized to the plasma membrane in non-TNBC cells and normal stem cells [91]. The intracellular CX26 in TNBC forms a tertiary complex with the activated form of focal adhesion kinase (p-FAK) and a pluripotency transcription factor NANOG. This CX26/p-FAK/NANOG complex drives self-renewal by activating NANOG translocation into the nucleus. On the other hand, in non-TNBC, membranous CX26 individually interacts with NANOG and p-FAK, and both proteins are retained at the membrane. Silencing CX26 in TNBC reduces the stability of the NANOG protein and activation of FAK, thus efficiently abrogating their tumor-initiating potential. Several chemicals, including carbenoxolone, linoleic acid, flufenamic acid, 2-aminoethoxydiphenyl borate, cyclodextrins, and diverse heterocyclic compounds, are known to inhibit CX26 (WO2004060398A1).

#### 2.4.2. Ubiquitin-Specific Protease (USP)

Ubiquitin-specific protease 2 (USP2) was first identified as a metastatic gene in TNBC by Qu et al. in 2014 [92]. In an attempt to identify predictive biomarkers for TNBC progression, they performed microarray analysis with primary TNBC tissue samples, including four from patients with metastasis and six from patients without metastasis. USP2 was identified to be up-regulated in patients with metastasis, and then upon further clinicopathological assessment, USP2 enrichment was detected in TNBC relative to that in non-TNBC [92]. He et al. found that silencing USP2 in TNBC reduced mammosphere formation and chemoresistance [93]. Their study showed that USP2 stabilized Twist by cleaving proteolytic ubiquitination, and Twist increased the transcription of Bmi1, which is a polycomb complex protein that controls the self-renewal and pluripotency of stem cells [93]. Davis et al. developed the USP2-specific inhibitor ML364, which directly binds and inhibits the enzymatic activity of USP2 [94]. ML364 efficiently inhibited the self-renewing function of TNBC cells and reduced tumor growth in vivo in xenograft models [93].

#### 2.4.3. Polo-Like Kinase (PLK)

Polo-like kinase 1 PLK1 is the best-characterized member of the human PLK family (PLK1-5) of serine/threonine protein kinases and is involved in various functions, such as mitotic entry, spindle assembly, and DNA damage response. Tan et al. revealed that PLK1 directly binds to MYC for its phosphorylation and induces MYC protein accumulation, in turn driving self-renewal activity and tumor-initiating potential [95,96]. Genomic expression profiling within multiple types of breast cancer indicated the overexpression of PLK1 in TNBC relative to that in other subtypes [97]. Interestingly, a PLK1 inhibitor, BI-2536, induces apoptosis in TNBC cells at the nanomolar range, while normal mammary epithelial cells remain intact at the same dose [97]. Moreover, BI-2536 treatment successfully ameliorated the CSC phenotype of TNBC cells and induced dramatic tumor growth inhibition in TNBC patient-derived tumor xenografts (PDXs) [95,97]. A phase II clinical trial of BI-2536 has been completed in recurrent and metastatic breast cancer patients (NCT00526149).

## 3. Attempts to Target Metabolic Reprogramming in Triple Negative Breast Cancer (TNBC)

Cancer stem cells (CSCs) possess the ability to survive in diverse microenvironments by obtaining energy from different sources depending on substrate availability. Indeed, accumulating evidence suggests that the preference of CSCs for glycolysis or oxidative phosphorylation (OXPHOS) is context-dependent [112]. Secondary pathways, such as fatty acid oxidation (FAO), serve as alternative strategies for fueling CSCs under additional energy-demanding conditions [112,113]. Moreover, metabolites from CSCs can affect nearby cell populations, such as T cells and macrophages, to help CSCs escape immune surveillance.

The metabolic phenotype of TNBC cells resembles that of CSCs. TNBC cells also take advantage of glycolysis over OXPHOS to more rapidly produce energetic and biosynthetic molecules even in the presence of sufficient O_2_. However, during chemotherapy, TNBC cells use OXPHOS to activate therapy resistance-associated signaling pathways [114]. Additionally, global metabolic profiling identified an increase in FAO intermediates in TNBC cells, and FAO was found to play a vital role in producing energy under metabolic stress to maintain TNBC survival [115,116,117]. In this section, we introduce the specific mechanisms regulating TNBC metabolism that are critical for TNBC survival and metastasis (Table 2).

### 3.1. Anaerobic Glycolysis

#### 3.1.1. Glycolysis in Breast Cancer Stem Cells (BCSCs)

Cancer stem cells (CSCs) take advantage of anaerobic glycolysis over OXPHOS even in the presence of O_2_, which is defined as the “Warburg effect”. OXPHOS requires mitochondrial complex activation and increases O_2_ consumption. Bypassing OXPHOS benefits CSCs by enabling rapid ATP synthesis and providing various types of metabolic intermediates for biosynthesis. Additionally, anaerobic glycolysis reduces the generation of reactive oxygen species (ROS) derived from the electron transport chain during OXPHOS. Accumulating evidence suggests that a set of glycolytic signatures, such as the elevation of glucose uptake, lactate production, and higher ATP content, are observed in BCSCs relative to those in bulk cancer cells [118,119,120]. In addition, the glycolytic inhibitor 2-deoxyglucose (2DG) can potently inhibit CD44+CD24- BCSCs by impairing mammosphere formation and in vivo tumorigenesis [121,122]. Moreover, glycolysis-mediated metabolites from BCSCs can affect various cellular compartments and thus alter the microenvironment to favor BCSCs. Lactate secretion may acidify the tumor microenvironment and favor the polarization of tumor-associated macrophages towards an M2 phenotype, which promotes proliferation, migration, and angiogenesis [123,124]. Furthermore, high rates of tumor glycolysis limit the availability of glucose for tumor-infiltrating lymphocytes, which require sufficient glucose for their effector functions, and impair T cell trafficking and T cell cytotoxicity [125,126].

#### 3.1.2. Glycolysis in TNBC Cells

Evidence suggests that compared to non-TNBC cells, TNBC cells exhibit a glycolytic signature. A quantitative analysis of the mitochondrial respiratory chain complex based on O_2_ consumption revealed that the activities of all segments of the respiratory chain in TNBC cell lines (MDA-MB-231 and MDA-MB-468) were lower than those in non-TNBC cell lines (BT474 and MCF7) [127]. The relative contribution of glycolysis and mitochondrial respiration to energy production was estimated in breast cancer cell line panels, including five TNBC and five non-TNBC cell lines [128]. Analysis of extracellular acidification rate (ECAR) and oxygen consumption rate (OCR) revealed that compared with non-TNBC cell lines, all TNBC cell lines displayed increased glycolysis and lactate production. Several key glycolytic proteins are preferentially overexpressed or activated in TNBC rather than in non-TNBC.

Hexokinase (HK) is an enzyme in the first step of glycolysis and catalyzes the conversion of glucose to glucose-6-phosphate (G-6-P). Lim et al. observed that TNBC patient tissues express HK2 at a higher level than non-TNBC patient tissues [128]. HK2 expression in TNBC was associated with epidermal growth factor receptor (EGFR) expression and required for EGF-induced lactate production, suggesting the essential role of HK2 in glycolysis [128]. The combinatorial inhibition of EGFR and glycolysis successfully reduced the in vitro and in vivo tumorigenesis of TNBC cells with minimal effect on non-TNBC cells [128].

Pyruvate kinase (PK) catalyzes the dephosphorylation of phosphoenolpyruvate to pyruvate, which is the last irreversible step of glycolysis. The M1 isoform of PK (PKM1) is expressed in most adult tissues, whereas the M2 isoform (PKM2) is expressed exclusively during embryonic development [129,130]. Interestingly, PKM2 is re-expressed in various types of tumor cells [131]. Notably, PKM2 exists in an active tetrameric complex in TNBC, whereas it appears in an inactive monomeric state in non-TNBC, suggesting that the switch from PKM1 to PKM2 provides a selective growth advantage in vivo. Through high-throughput screening, Heiden et al. recently discovered the hit compound for PKM2 inhibitors; however, their clinical use is not warranted because of the limited therapeutic window [114]. Dong and coworkers provided a novel mechanism of PKM2 activation in TNBC [132]. In TNBC patient tissues and cell lines, Snail epigenetically silenced fructose-1,6-biphosphatase (FBP1) expression [132]. This Snail-mediated loss of FBP1 increased the concentration of fructose-1,6-bisphosphate, which activated PKM2 enzyme activity, resulting in the elevation of glycolytic flux and ATP production to maintain TNBC cells even under hypoxia [132]. Moreover, PKM2-mediated glycolysis was required for TNBC cells to sustain the CD44+CD24- phenotype and BCSC functions, including mammosphere formation and in vivo tumorigenesis [132].

Pyruvate dehydrogenase kinase 1 (PDK1) phosphorylates the pyruvate dehydrogenase (PDH) E1α subunit and inactivates the PDH enzyme complex that converts pyruvate to acetyl-coenzyme A (acetyl-CoA), thereby inhibiting pyruvate oxidation via the tricarboxylic acid cycle to generate energy. Peng et al. discovered that PDK1 was elevated in CD44+CD24- BCSCs versus CD44-CD24+ counterparts [133]. Additionally, PDK1 protein and mRNA levels were elevated in MBA-MB-231 mammospheres and MDA-MB-231 cell subpopulations with high ALDH activity [133]. Genetic silencing of PDK1 has been shown to significantly reduce mammosphere formation and in vivo tumorigenesis by MDA-MB-231 cells [133]. The study demonstrated that PDK1 transcription was dependent on HIF1α and that PDK1 was involved in glycolysis activation under hypoxic conditions to increase stemness in MDA-MB-231 cells [133].

#### 3.1.3. Glycolysis Inhibitors in Clinical Trials

Although the number of glycolysis inhibitors is fairly limited, the widely used anti-diabetic drug metformin has been proposed as a potent anticancer drug due to its effects on multiple signaling pathways, including the mammalian target of rapamycin complex 1 (mTORC1), AMP-activated protein kinase, and Rag GTPase [134,135]. Indeed, recent experimental results have provided new insights into the mechanisms of action of metformin as a key regulator of cellular metabolism, including decreased energy production and inhibited glucose uptake [136,137]. Moreover, metformin directly binds to HK2 to inhibit its enzymatic activity and induces the dissociation of HK2 from the mitochondria [138]. Metformin is currently being evaluated in phase III clinical trials in patients with diverse cancer types, including TNBC (NCT02201381). A series of novel metformin derivatives, such as HL010183, have been synthesized to improve antitumor activity [139].

Notably, Li et al. recently adopted the structure-based virtual ligand screening method to screen the FDA-approved drug database and identified benserazide as an HK2 inhibitor [140]. Benserazide has been used in combination with levodopa for the treatment of Parkinson’s disease. Its definite pharmacokinetics, pharmacodynamics and low toxicity have largely encouraged the repurposing of benserazide and its derivatives as antitumor agents.

TLN-232 is a seven-amino acid peptide that inhibits PKM2 activity, and it has been evaluated in a phase II clinical trial in patients with refractory metastatic renal cell carcinoma (NCT00422786). However, a phase II trial in metastatic melanoma patients was halted for legal reasons (NCT00735332).

The PDK1 inhibitor dichloroacetate (DCA) is effective in many cancer types, including colon cancer, breast cancer and oral squamous cell carcinoma [141,142,143]. In cancer cells, DCA switches glucose metabolism from aerobic glycolysis to glucose oxidation. DCA increases OXPHOS and ROS production in mitochondria, which limits proliferation and increases apoptosis in cancer cells. The phase II study evaluating the use of DCA in metastatic breast cancer and lung cancer patients was terminated due to higher than expected risk/safety concerns (NCT01029925). AR-12 has been developed as an orally bioavailable small-molecule inhibitor for PDK1, and a phase I study of AR-12 has been completed in patients with advanced or recurrent solid tumors or lymphoma (NCT00978523).

### 3.2. OXPHOS

#### 3.2.1. OXPHOS in BCSCs

Luo and coworkers recently shed new light on OXPHOS in BCSCs [122]. They showed that mitochondria in BCSCs do not definitely lose their ability to carry out OXPHOS. Instead, under conditions of glycolysis inhibition, BCSCs are reprogrammed to use OXPHOS with higher ROS levels and become more reliant on antioxidant responses [122]. Conversely, by reducing ROS levels using ROS scavengers, BCSCs can shift towards a more glycolytic phenotype instead of undergoing OXPHOS [122]. Emerging evidence shows that BCSCs can acquire a hybrid glycolysis/OXPHOS phenotype in which both glycolysis and OXPHOS can be used for energy production and biomass synthesis. The hybrid glycolysis/OXPHOS phenotype facilitates the metabolic plasticity of BCSCs and may be specifically associated with metastasis and therapy resistance [144].

#### 3.2.2. OXPHOS in TNBC Cells

Recently, the use of OXPHOS by TNBC cells was observed during the development of therapy resistance, which is linked to CSC phenotype enrichment in the residual tumor after chemotherapy. Lee et al. discovered that TNBC cells increase mitochondrial OXPHOS after chemotherapy, resulting in ROS production [145]. Subsequently, ROS augment HIF1α stabilization, which in turn promotes cancer stemness and chemotherapy resistance [145]. Bhola et al. also revealed a possible relationship between OXPHOS and the intrinsic resistance of TNBC cells [146]. The phosphoinositide 3-kinase (PI3K) and mammalian target of rapamycin (mTOR) signaling pathways are highly activated in TNBC cells; however, PI3K/mTOR inhibitors and mTORC1/2 inhibitors have shown only limited success because of resistance [146]. These resistant TNBC cells exhibit stem-like properties, such as CD44 and ALDH positivity and mammosphere formation. TNBC cells were discovered to activate the Notch signaling pathway following mTORC1/2 inhibition by increasing the expression of Notch1, JAG1, and active Notch1 intracellular domain (NICD) [146]. This Notch1 signaling activation is dependent on mitochondrial OXPHOS via the up-regulation of mitochondrial transcription factor A and the ATP synthase complex subunits ATP5G2 and ATP5J2 [146].

#### 3.2.3. OXPHOS Inhibitors

Despite the importance of OXPHOS, adequate therapeutic strategies are not currently available. Recently, Molina et al. developed a specific small-molecule inhibitor for mitochondrial electron transport chain complex I, called IACS-010759. Its therapeutic effects are being investigated in multiple types of cancer, and it has shown considerable growth inhibition in thirteen of the sixteen TNBC cell lines studied [147,148]. A phase I clinical trial is ongoing in patients with advanced cancer, including TNBC (NCT03291938). ME-344 is another mitochondrial complex I inhibitor that is currently being evaluated in a phase I clinical trial in breast cancer patients (NCT02806817).

### 3.3. FAO

#### 3.3.1. FAO in BCSCs

To catabolize long-chain fatty acids to acetyl-CoA, long-chain acyl-CoA synthetase (LACS) converts fatty acids to fatty acyl-coenzyme A (acyl-CoA) at the mitochondrial outer membrane. Acyl-CoA must be enzymatically converted to acylcarnitine by carnitine palmitoyltransferase 1 (CPT1A or CPT1B) to cross the mitochondrial inner membrane and enter the FAO cycle. Thus, acylcarnitine is an essential intermediate and is involved in the first step of FAO.

Wang et al. recently compared the rate of FAO in BCSCs versus non-BCSCs by estimating the oxidation rate of ^3^H-palmitic acid. The production of ^3^H_2_O was significantly higher in patient-derived mammospheres than in bulk tumor cells [117]. The same phenomenon was also observed in CD44+CD24- BCSCs isolated from MDA-MB-468 cells, suggesting that the FAO rate is elevated in BCSCs relative to that in non-BCSCs [117]. Moreover, the reduction in FAO by CPT1 inhibition preferentially diminishes ATP production in BCSCs without affecting non-BCSCs, leading to a decrease in the BCSC population and tumor-initiating potential [117]. This study proposed the importance of breast adipocytes for FAO activation in BCSCs. Adipocytes near BCSCs secrete leptin, which binds to LEPR on BCSCs, leading to JAK/STAT3 signaling activation. Activated STAT3 binds to the CPT1B promoter to increase its transcription, and FAO is activated in BCSCs, suggesting that the LEPR/JAK/STAT3 pathway is an FAO-activating mechanism in BCSCs.

#### 3.3.2. FAO in TNBC Cells

The results of metabolomics analyses conducted by Wang et al. revealed that the levels of long-chain FAO metabolites (acylcarnitines) were higher in TNBC cell lines (MDA-MB-436 and Hs578T) than in non-TNBC cell lines (HCC1500 and BT20) [117]. Camarda et al. performed global metabolic analyses using mass spectrometry and reported that all six acylcarnitine intermediates were significantly elevated in TNBC cells in correlation with MYC expression [116]. The same group found an increase in a set of genes that encode activators of FAO in TNBC patients versus non-TNBC patients [116]. Additionally, FAO inhibition by CPT1 knockdown preferentially reduced ATP production and growth in MYC-overexpressing TNBC cells, with minimum effects on non-TNBC cells [116].

Wright and coworkers recently reported that the cell surface antigen CUB domain-containing protein 1 (CDCP1) was correlated with the lipid metabolism pathway in TNBC and that CDCP1 contributed to the energy production required for the migration and metastasis of TNBC [149]. Their study demonstrated that CDCP1 interacts with LACS to stimulate FAO; in turn, excessive acetyl-CoA production increases OXPHOS to produce ATP. Blocking CDCP1 activation by overexpressing the secreted extracellular portion of cleaved CDCP1 reduces tumor growth and metastasis in TNBC cell-derived xenografts (MDA-MB-231 and UCI-082014) [149].

FAO is also linked to chemoresistance. Paclitaxel-resistant MDA-MB-231 cells, which were generated by serial passage in medium supplemented with paclitaxel, displayed increased mRNA levels of CPT1B and the FAO enzyme ACADM as well as increases in stem cell marker expression (MSI1 and OCT4), mammosphere formation and FAO [117]. In addition, the transfer of long-chain fatty acids into the mitochondrial matrix for FAO was promptly activated following paclitaxel treatment in paclitaxel-resistant MDA-MB-231 cells but not in parental MDA-MB-231 cells. CPT1 inhibition sensitized these resistant MDA-MB-231 cells to paclitaxel [149].

#### 3.3.3. FAO Inhibitors in Clinical Trials

Etomoxir is a widely used small-molecule inhibitor of FAO because of its irreversible inhibitory effects on CPT1. Etomoxir treatment can significantly reduce the viability of BCSCs with high CPT1 expression and high FAO rates without affecting non-BCSCs with low FAO rates and minimal CPT1 expression [117]. Moreover, etomoxir treatment significantly reduces tumorigenesis in TNBC mouse models, including FVB/N allografts and HCI-002 xenografts, with an increase in survival rates [116]. Perhexiline is another small-molecule inhibitor of CPT1, and perhexiline treatment reduces the proportion of CD44+/Sca-1+ BCSCs in tumors harvested from MMTB-PyMT mice with the reduction in SOX2 and ALDH1A expression [117]. A double-blind randomized clinical trial (the ERGO study) was performed to evaluate the efficacy and safety of etomoxir in patients with congestive heart failure [150]. Perhexiline has also been tested in phase II and III trials as an antianginal agent to increase myocardial efficiency (NCT00845364). However, none of these FAO inhibitors have been tested as anticancer agents in the clinic.

### 3.4. Other Molecules Linked to TNBC Metabolism

#### Glutathione S-Transferase Pi 1 (GSTP1)

In an attempt to identify TNBC-specific metabolic enzyme targets, Louie et al. compared a panel of four non-TNBC cell lines (MCF7, T47D, ZR751, and MDA-MB-361) and five TNBC cell lines (231 MFP, HCC1143, HCC38, HCC70, and MDA-MB-468) using chemoproteomic profiling [151]. GSTP1 was the most significantly up-regulated target in TNBC cells, and it was found to interact with GAPDH to enhance GAPDH activity [151]. Upon GSTP1 knockdown, glycolytic flux was inhibited in TNBC cells, along with a significant reduction in lactic acid secretion and glucose consumption [152]. LAS17 was recently developed as a GSTP1 inhibitor that targets active-site tyrosine [151]. Treatment with LAS17 reduced the survival potential of TNBC cells under serum-free conditions and inhibited tumorigenesis in 231 MFP xenografts [152].

## 4. Conclusions and Future Perspectives

TNBC has an aggressive nature, and patients diagnosed with TNBC lack targeted therapy options. Thus, the development of potential therapeutic targets is urgently needed. In this review, we have discussed the molecular mechanisms shared by both TNBC cells and CSCs with a focus on SRSPs and metabolic reprogramming (Figure 2).

The activity of SRSPs of TNBC is highly dependent on several key mechanisms, such as STAT3, SRC kinase, and Wnt/β-catenin signaling. Many inhibitors targeting these signaling pathways are currently undergoing preclinical and clinical development. Cellular metabolism in TNBC is highly dependent on glycolysis. OXPHOS plays a critical role in the development of therapy resistance in TNBC, suggesting that TNBC cells exhibit metabolic plasticity that is similar to CSCs. Moreover, FAO is involved in the maintenance of CSCs and chemoresistance of TNBC cells by compensating for the high metabolic demands. Some metabolic inhibitors are being investigated and have shown significant inhibitory effects on multiple cancer types, including TNBC. Nevertheless, the number of metabolic inhibitors is relatively limited; thus, future investigations should focus on revealing the detailed mechanisms underlying the metabolic plasticity of TNBC.

Considering the complexity and diversity of TNBC, eradicating all cancer cells using only one strategy is a challenging task. However, many drugs targeting cancer stemness have shown a significant effect on TNBC. Thus, the use of combinatorial treatments comprising CSC-targeting agents and conventional chemotherapeutic drugs is a promising strategy to induce synergistic effects against TNBC.

## Figures and Tables

**Figure 1 cancers-11-00965-f001:**
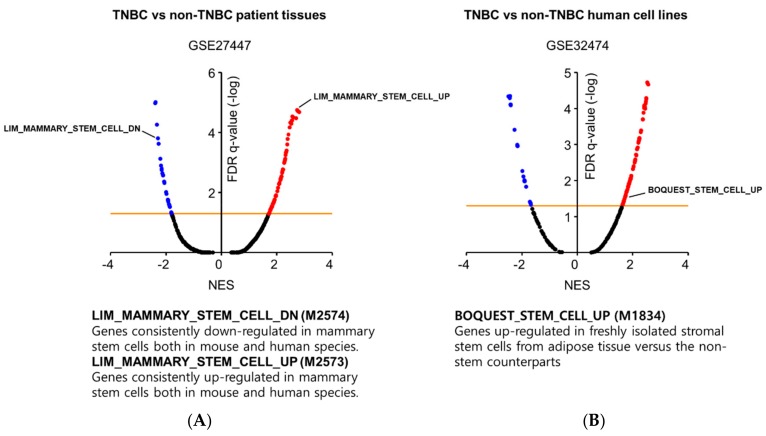
GSEA results showing significant enrichment of stem cell gene sets in (**A**) TNBC patient tissues and (**B**) TNBC human cell lines in relation to non-TNBC counterparts. Gene sets are ordered by the normalized enrichment score (NES); the orange line indicates statistical significance (FDR q-value < 0.05).

**Figure 2 cancers-11-00965-f002:**
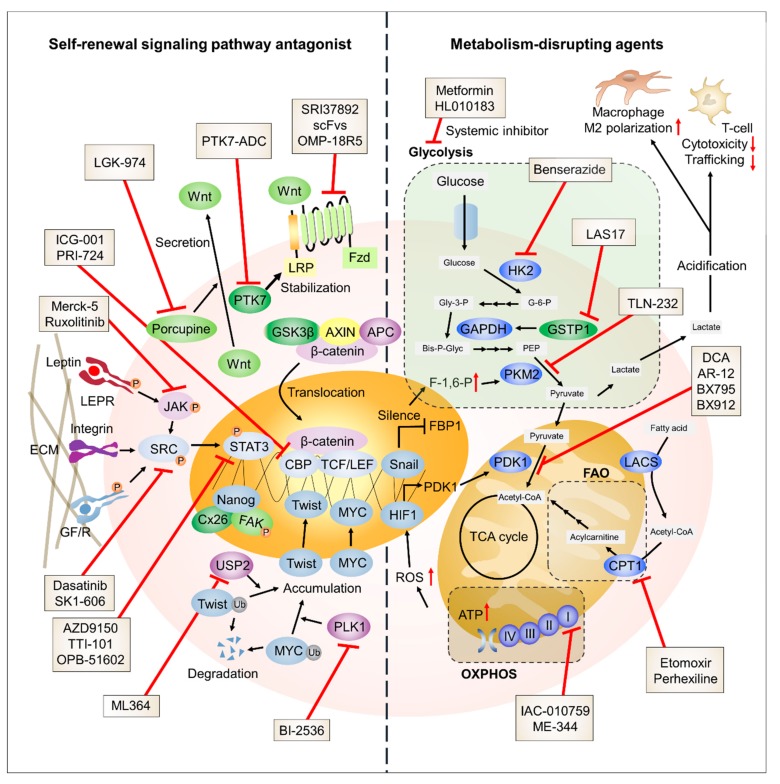
Schematic representation of TNBC-targeting strategies. The potential therapeutic targets involved in SRSPs and metabolic processes are presented with their specific inhibitors.

**Table 1 cancers-11-00965-t001:** Therapeutic attempts to inhibit the self-renewal process of TNBC cells.

Target	Drug	Preclinical Results	Clinical Trial Status and Results
STAT3 signaling pathway inhibitors			
JAK	Merck-5 (JAK inhibitor I)	It reduces the growth of basal-like tumor cells through the inhibition of STAT3 activity in vitro [39].	Preclinical
	Ruxolitinib	It reduces the proliferation, invasion, and mammosphere formation in HCC38 cells and increases apoptosis [98].	Phase II -NCT01562873: The trial was terminated because of the limited responses to continue treatment despite on-target activity in refractory, metastatic TNBC patients [99]. -NCT02876302: The trial is currently recruiting patients to test the combinatory effect of ruxolitinib with paclitaxel on triple-negative inflammatory breast cancer patients.
STAT3	OPB-51602	It reduces mammosphere formation and CD44+/CD24− BCSC populations in MDA-MB-231 cells [100].	Phase I -NCT01184807: The trial has been completed in patients with malignant solid cancer, and OPB-51602 demonstrated promising antitumor activity, particularly in non-small-cell lung cancer. It has a long half-life and poor tolerability to continuous dosing compared with intermittent dosing [101].
	AZD9150	It shows anti-proliferative efficacy as a single-agent in lymphoma and lung cancer preclinical mouse models. Its clinical trial is currently recruiting patients with advanced solid tumors [102].	Phase I and II -NCT03421353: The trial is currently recruiting patients with advanced cancer. Phase I/Ib -NCT01839604: The trial has been completed in patients with advanced/metastatic hepatocellular carcinoma.
	TTI-101 (STAT3 inhibitor XIII)	It reduces the in vitro cell proliferation more potently in TNBC cells (MDA-MB-468 and MDA-MB-231) than in non-TNBC cells (MDA-MB-435 and MCF7). This clinical trial is currently recruiting patients with advanced cancers, including breast cancer [103].	Phase I -NCT03195699: The trial is currently recruiting advanced cancer patients.
SRC kinase signaling pathway inhibitors			
SRC	Dasatinib	It reduces the proliferation of TNBC cells in vitro and their tumorigenic potential in vivo. It sensitizes TNBC cells to paclitaxel [41,61].	Phase II -NCT02720185: The trial was planned for the study of TNBC patients with nuclear translocation of EGFR; however, the trial has been suspended for protocol modifications.
	SKI-606 (Bosutinib)	It reduces tumor growth, invasion, and metastasis in MDA-MB-231 xenografts [62].	Phase I -NCT03854093: The trial is currently recruiting breast cancer patients. -NCT03023319: The trial is currently recruiting patients with metastatic solid cancers to assess SKI-606 in combination with pemetrexed. -NCT02810990: The trial is currently recruiting elderly chronic myeloid leukemia patients.
Wnt/β-catenin signaling inhibitors			
CBP	ICG-001	It reduces mammosphere formation and sensitizes TNBC cells to paclitaxel [75].	Preclinical
	PRI-724	It is an ICG-001 derivative for clinical trials in patients with pancreatic cancer and myeloid leukemia [104].	Phase I and II -NCT01606579: The trial has been completed in subjects with advanced myeloid malignancies, but the results have not yet been reported. Phase I -NCT01764477: The trial examined patients with advanced or metastatic pancreatic adenocarcinoma to assess PRI-724 in combination with gemcitabine (GEM), and the results show that this combination is safe and demonstrates modest clinical activity [105].
Porcupine	LGK-974	It reduces tumor growth in metastatic MDA-MB-231 cell (TMD-231) xenografts. It sensitizes TMD-231 cells to buparlisib [106].	Phase I -NCT01351103: The trial is currently recruiting patients with malignancies dependent on Wnt ligands.
FZD7	SRI37892	It reduces tumor growth and tumor-initiating potential in TNBC patient tissue- and cell line-derived xenografts [107].	Preclinical
	scFvs	It inhibits cell growth inhibition and promotes apoptosis in MDA-MB-231 cells without affecting SK-BR3 cells [108,109].	Preclinical
	OMP-18R5 (vantictumab)	It promotes tumor growth regression by Taxol and prevents recurrent growth after Taxol treatment in breast cancer patient tissue-derived xenografts [89].	Phase Ib -NCT01973309: The trial has been completed in patients with recurrent or metastatic breast cancer to evaluate OMP-18R5 in combination with paclitaxel. This combination was demonstrated to be well tolerated. Bone toxicity was encountered early in the study [110].
PTK7	PTK7-ADC	It reduces tumor growth and tumor-initiating potential in TNBC patient tissue- and cell line-derived xenografts [86].	Phase I -NCT03243331: The trial is currently recruiting metastatic TNBC patients to assess a combination drug regimen with gedatolisib.
Other molecules linked to the self-renewal process in TNBC			
CX26	-	The specific inhibitors have not been developed yet.	-
USP2	ML364	It reduces tumorsphere formation in vitro and tumor growth in vivo. It sensitizes TNBC cells to doxorubicin and paclitaxel [93].	Preclinical
PLK1	BI-2536	It reduces tumor growth in TNBC xenografts [97].	Phase II -NCT00526149: The trial has been completed in patients with recurrent or metastatic solid cancer, and BI-2536 showed limited antitumor activity [111].

**Table 2 cancers-11-00965-t002:** Therapeutic attempts to target TNBC metabolism.

Target	Drug	Preclinical Results	Clinical Trial Status and Results
Glycolysis inhibitors			
HK2	Metformin	A systemic glycolysis inhibitor; it suppresses TNBC stem cells and reduces the tumor-initiating potential in TNBC xenografts [137,153].	Phase III -NCT02201381: The trial is currently recruiting cancer patients; overall survival is the primary outcome measure.
	HL010183	A metformin derivative; it inhibits proliferation and invasion of TNBC cells and reduces tumor growth in MDA-MB-231 xenografts [139].	Preclinical
	Benserazide	FDA-approved drug for Parkinson’s disease. It reduces anaerobic glycolysis in breast cancer cells and inhibits tumor growth [140].	Preclinical
PKM2	TLN-232	It has anti-proliferative effects on diverse cancer cells [154,155].	Phase II -NCT00422786: The trial has been completed in patients with refractory metastatic renal cell carcinoma, but the results have not yet been reported. -NCT00735332): The trial was conducted in recurring metastatic melanoma patients, but it was stopped because of license termination.
PDK1	DCA	It inhibits metastatic breast cancer cell growth in vitro and in vivo [142].	Phase II -NCT01029925: The trial was conducted in patients with metastatic breast cancer or with lung cancer, but it was terminated early due to higher than expected risk/safety concerns.
	AR-12 (OSU-03012)	It reduces proliferation and induces apoptosis of MDA-MB-231 cells in vitro and in vivo. Additionally, it sensitizes MDA-MB-231 cells to tamoxifen [156].	Phase I -NCT00978523: The trial was conducted in patients with advanced or recurrent solid tumors or with lymphoma.
	BX795/BX912	It reduces the cell viability of MYC-expressing TNBC cells (MDA-MB-231, SUM159PT, Hs578T) but does not affect non-TNBC cells (BT474 and T47D). In addition, it attenuates the CD44+/CD24- population in MDA-MB-231 cells [95,157].	Preclinical
OXPHOS inhibitors			
Mitochondrial complex I	IACS-010759	It reduces cell growth and viability across a panel of cancer cell lines, including TNBC without affecting normal cells [147].	Phase I -NCT03291938: The trial is currently recruiting patients with advanced cancer. -NCT02882321: The trial is currently recruiting subjects with relapsed or refractory acute myeloid leukemia.
	ME-344	It sensitizes breast tumors to tyrosine kinase inhibitors in mouse mammary tumor virus-polyoma middle tumor-antigen (MMTV-PyMT) mouse model [158,159,160].	Phase I -NCT02100007: The trial was terminated because of the lack of efficacy in solid tumor patients as a combinatory agent with Hycamtin^®^ -NCT01544322: The trial was completed in patients with refractory solid cancer; however, the results have not yet been released. -NCT02806817: The trial was recruiting HER2-negative breast cancer patients with antiangiogenic-induced mitochondrial metabolism, but the trial status has not been verified in over two years.
FAO inhibitors			
CPT1	Etomoxir	It reduces the ATP production in MYC-expressing TNBC cells, thus leads to tumor regression in vitro and in vivo. Moreover, it reduces the CSC proliferation and their self-renewing activity in TNBC cells [116,117].	Preclinical
	Perhexiline	It reduces tumor growth, CSC population, and Sox2 expression in MMTV-PyMT tumors. Additionally, it restores the efficacy of paclitaxel in the paclitaxel-resistant MDA-MB-231 cells [117].	Phase II and III -NCT00845364: The trial was conducted to assess whether an anti-anginal agent could protect the myocardium in patients undergoing coronary artery surgery (CASPER). The role of perhexiline in cardiac surgery is limited [161].
Other molecules linked to TNBC metabolism			
GSTP1	LAS17	It reduces survival in TNBC cells and tumor growth in TNBC xenografts [152].	Preclinical

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
