# Peer review of "Targeting Cancer Stem Cells in Triple-Negative Breast Cancer"

_cancers, 2019, doi:10.3390/cancers11070965_

Round 1

Reviewer 1 Report

This manuscript by Park S et al., provides a comprehensive review of current knowledge on how triple negative breast cancer (TNBC) cells harbor cancer stem cell (CST) signatures and presents directions for future research in the field. The outline of the review article is rational and appropriate, including a succinct introduction, approaches for targeting the self-renewal process in TNBC, strategies to target metabolic reprogramming in TNBC, ending with a brief conclusion and future directions. The review highlights important preclinical and clinical studies in the field.  The figures and tables which complement the manuscript are well utilized and described appropriately. The references are accurate. The manuscript is timely and should capture the interest of the wide TNBC audience.

A few concerns should be addressed:

1.       A list of abbreviations and the full meanings should be provided.

2.       Recent studies suggest that simultaneous inhibition of IL-6 and IL-8 is required to induce stem cell apoptosis to alter their propensity for TNBC tumor initiation 9hartman ZC et al., Cancer Res, 2013, 73: 3470-3480). However, it is unclear why the authors dis not mention the role of IL-8 in the activation of STAT-3 signaling in cancer stem cell (CSC) in TNBC.

3.       Authors should discuss the role or lack thereof of STAT-5 in TNBC CSC.

4.       In Sections 3.1.1 and 3.1.2: change O2 to O2.

Author Response

Please find enclosed our revised manuscript (ID: cancers-535598) titled "Targeting Cancer Stem Cells in Triple-Negative Breast Cancer" by So-Yeon Park et al. to be considered for publication in Cancers. We are truly glad that our manuscript has been improved by your insightful comments. We addressed all of your concerns, hoping for your agreement. Thank you in advance for taking your time and effort for our review article, and we look forward to hearing from you soon.

Reviewer 2 Report

This paper by Park et al is a comprehensive review of stem cell features observed in TNBC, with putative therapeutic strategies adapted to these biological characteristics.

They make an exhaustive description of preclinical data (as represented in the Figure 2 that must be kept in future versions of this review).

Nevertheless, the clinical part of the paper should be improved by detailing the clinical results of already published early phase trials mentioned in the text. Moreover, some trials that have been prematurely closed due to futility but have never been published should also be described. This would add insights concerning the clinical utility (efficacy and safety) of some drugs that have not reached the final steps of drug development. To be consistent with this comment, tables should be completed with known clinical results of early phase clinical trials.

I think that unpublished author's data described in Page 2, paragraph 3, and table 1 have not been peer reviewed and validated and thus should not be mentioned in this review.  

There are some typos in the text. For instance, in Page 3, 2nd paragraph, "we introduced" should be replaced by "we introduce".

Author Response

(The authors gave the same response as above.)

Round 2

Reviewer 2 Report

Dear author,

I'd like to thank you for the additional data related to clinical results and English editing.

Hope it will be enough to see your apper published.

Sincerely yours